# ACE: Artificial Checkerboard Enhancer to Induce and Evade Adversarial Attacks

## Abstract

The checkerboard phenomenon is one of the well-known visual artifacts in the computer vision field. The origins and solutions of checkerboard artifacts in the pixel space have been studied for a long time, but their effects on the gradient space have rarely been investigated. In this paper, we revisit the checkerboard artifacts in the gradient space which turn out to be the weak point of a network architecture. We explore image-agnostic property of gradient checkerboard artifacts and propose a simple yet effective defense method by utilizing the artifacts. We introduce our defense module, dubbed Artificial Checkerboard Enhancer (ACE), which induces adversarial attacks on designated pixels. This enables the model to deflect attacks by shifting only a single pixel in the image with a remarkable defense rate. We provide extensive experiments to support the effectiveness of our work for various attack scenarios using the state-of-the-art attack methods. Furthermore, we show that ACE is even applicable to large-scale datasets including ImageNet dataset and can be easily transferred to various pretrained networks.

## 1 Introduction

The checkerboard phenomenon is one of the well-known artifacts that arise in various applications such as image super resolution, generation and segmentation (Shi et al., 2016; Gao et al., 2018; Sajjadi et al., 2017). In general, the checkerboard artifacts indicate an uneven pattern on the output of deep neural networks (DNNs) that occurs during the feed-forward step. Odena et al. (2016) have investigated in-depth the origin of the phenomenon that the artifacts come from the uneven overlap of the deconvolution operations (i.e., transposed convolution, Dumoulin et al. (2016)) on pixels. Its solutions have been suggested in various studies (Wojna et al., 2017; Aitken et al., 2017).

Interestingly, a possible relationship between the checkerboard artifacts and the robustness of neural networks has been noted by Odena et al. (2016) but not has it been seriously investigated. Moreover, while the previous works (Long et al., 2015; Noh et al., 2015; Odena et al., 2016; Feinman et al., 2017) have concentrated on the artifacts in the *pixel space*, studies have been rare on the artifacts in the *gradient space* that occur during the backward pass of the convolution operation.

To show that the gradient checkerboard artifacts phenomenon is crucial for investigating the network robustness and is indeed a weak point of a neural network, we focus on analyzing its effects on the gradient space in terms of adversarial attack and defense. By explicitly visualizing the gradients, we demonstrate that the phenomenon is inherent in many contemporary network architectures such as ResNet (He et al., 2016), which use strided convolutions with uneven overlap. It turns out that the gradient checkerboard artifacts substantially influence the shape of the loss surface, and the effect is image-agnostic.

Based on the analysis, we propose an *Artificial Checkerboard Enhancer* module, dubbed ACE. This module further boosts or creates the checkerboard artifacts in the target network and manipulates the gradients to be caged in the designated area. Because ACE guides the attacks to the intended environment, the defender can easily dodge the attack by shifting a single pixel (Figure 1) with a negligible accuracy loss. Moreover, we demonstrate that our module is scalable to large-scale datasets such as ImageNet (Deng et al., 2009) and also transferable to other models in a plug and play fashion without additional fine-tuning of pretrained networks. Therefore, our module is highly practical in general scenarios in that we can easily plug any pretrained ACE module into the target architecture.

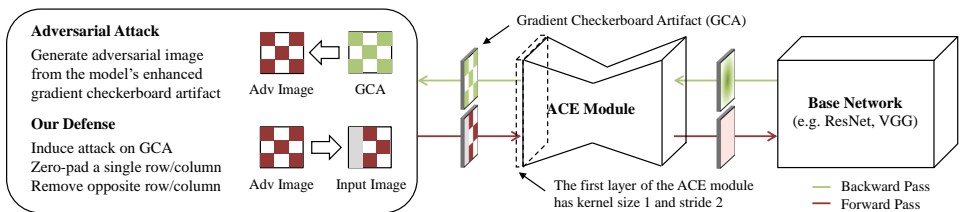

Figure 1: Defense procedure using the proposed Artificial Checkerboard Enhancer (ACE) module. ACE shapes the gradient into a checkerboard pattern, thus attracting adversarial attacks to the checkerboard artifacts. Since the defender is aware of the guided location in advance, adversarial attacks can be easily deflected during inference by padding the image with a single row/column and discarding the opposite row/column.

Our contributions are summarized as three-fold:

**Analysis of gradient checkerboard artifacts.** We investigate the gradient checkerboard artifacts in depth which are inherent in many of the contemporary network architectures with the uneven overlap of convolutions. To the best of our knowledge, this is the first attempt to analyze the artifacts of the gradient space in terms of network robustness. We empirically show that the gradient checkerboard artifacts incur vulnerability to the network.

**Artificial Checkerboard Enhancer (ACE).** We introduce ACE module that strengthens gradient artifacts and induces adversarial attacks to our intended spaces. After guiding the attacks to the pre-specified area using ACE, we deflect adversarial attacks by one-pixel padding. Our extensive experimental results support that our proposed defense mechanism using ACE module successfully defends various adversarial attacks on CIFAR-10 (Krizhevsky, 2009) and ImageNet (Deng et al., 2009) datasets.

**Scalability.** We show that ACE is readily transferable to any pretrained model without fine-tuning, which makes the module scalable to a large-scale dataset. To the best of our knowledge, this is the first defense method that attempts and succeeds to defend the attacks with the projected gradient descent algorithm (PGD) (Madry et al., 2017; Athalye & Sutskever, 2017) on ImageNet dataset.

## 2 PRELIMINARY

### 2.1 BACKGROUND AND RELATED WORKS

Adversarial attacks can be conducted in various ways depending on how much the adversary, also known as the threat model, has access to the target model. If the attacker can acquire the gradients of the target, gradient-based attacks can be performed, which are usually very effective because iterative optimization becomes possible (Goodfellow et al., 2014; Kurakin et al., 2016; Moosavi-Dezfooli et al., 2016; Papernot et al., 2016a; Carlini & Wagner, 2016). Score-based attacks can be valid when the adversary can use the logits or the predicted probabilities for generating adversarial examples (Su et al., 2017). If generated adversarial examples are not from the target model, we call this transfer-based attack (Papernot et al., 2017). Recently, a new type of attack, called a decision-based attack, has been introduced where the adversary only has knowledge about the final decision of the model (e.g., top-1 class label) (Brendel et al., 2017).

According to Papernot et al. (2017), defense methods can be largely categorized into gradient masking and adversarial training. Gradient masking methods usually make adversaries difficult to compute exact gradients and make it challenging to fool the target. Gradient obfuscation, which is a recently introduced term of gradient masking by Athalye et al. (2018), includes specific gradient categories such as stochastic gradients, shattered gradients and vanishing/exploding gradients. Works related to our defense method can be considered as input transformations which focus on the input image (Xie et al., 2017; Prakash et al., 2018; Cao & Gong, 2017; Xu et al., 2017; Guo et al., 2017).

On the other hand, adversarial training has been known to make models robust against adversarial perturbation (Madry et al., 2017; Na et al., 2017; Tramèr et al., 2017), but there remains an issue

that the robustness comes with the cost of accuracy (Tsipras et al., 2018; Su et al., 2018). Moreover, according to Sharma & Chen (2018), restricting on $l_\infty$ bounded perturbations during adversarial training has limited robustness to attacks with different distortion metrics.

## 2.2 NOTATION

We would like to define the following terms here and will use them without further explanation throughout this paper. **Gradient Overlap** ($\Omega(\boldsymbol{x}_i)$) represents the number of parameters associated with a single pixel in the input $\boldsymbol{x}$. For more explanation on its calculation, see Appendix C. We define the set of pixels whose gradient overlap is in the top $p$ fraction as $\mathcal{G}(p)$ (e.g., $\mathcal{G}(1.0)$ represents the entire pixel set). **Gradient Checkerboard Artifacts (GCA)** is a phenomenon which shows checkerboard patterns in gradients. The existence of GCA has been introduced in Odena et al. (2016), although not has it been thoroughly examined. GCA occurs when a model uses a convolution operation of kernel size $k$ that is not divisible by its stride $s$.

# 3 REVISITING GRADIENT CHECKERBOARD ARTIFACT IN TERMS OF GRADIENT OVERLAP

## 3.1 MOTIVATION

We first introduce a simple experiment that motivated the design of our proposed ACE module. We conduct an experiment to visualize the attack success rate of a given image set using a single pixel perturbation attack. For each pixel of an input image, we perturb the pixel to white (i.e., a pixel with RGB value of (255, 255, 255)). Next, we use our toy model based on LeNet (LeCun et al., 1998) (see Appendix A for the detailed architecture) and ResNet-18 (He et al., 2016) to measure the attack success rate on the test images in CIFAR-10 dataset. Note that the attack success rate $P_{\text{attack}}$ in Figure 2 is defined as the average number of successful attacks per pixel on the entire set of test images.

As we can see in Figure 2a and Figure 2c, checkerboard patterns in the attack success rate are clearly observable. This pattern can be considered as image-agnostic because it is the result of the average over the entire set of the test images of CIFAR-10 dataset. Then a natural question arises: **What is the cause of this image-agnostic phenomenon?** We speculate that the uneven gradient overlap is the cause of this phenomenon, which is directly associated with the number of parameters that are connected to a single pixel. As depicted in Figure 2b and Figure 2d, we can observe checkerboard patterns in the gradient overlap. In fact, this uneven overlap turns out to be substantially susceptible to the adversarial attacks. We will provide the supporting results on this in the following sections.

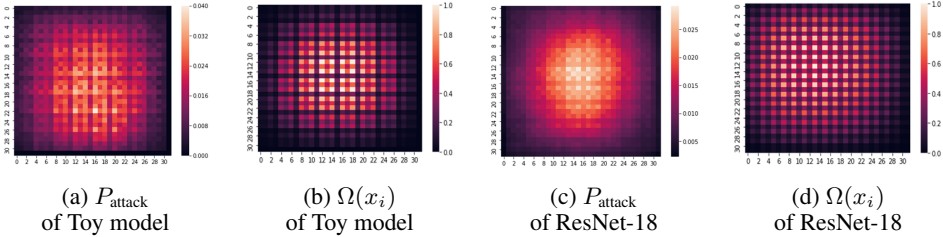

|   (a) $P_{\text{attack}}$   |   (b) $\Omega(x_i)$   |   (c) $P_{\text{attack}}$   |   (d) $\Omega(x_i)$   |
| :---: | :---: | :---: | :---: |
| of Toy model | of Toy model | of ResNet-18 | of ResNet-18 |

Figure 2: Illustration of the attack success rate $P_{\text{attack}}$ and the gradient overlap $\Omega(x_i)$ of toy model and ResNet-18. The illustrated gradient overlap of ResNet-18 comes from the features after the fifth convolutional layer. Attack success rate (a) and (c) are computed by perturbing each pixel of an image to white (i.e., (255, 255, 255)), over the entire set of test images in CIFAR-10 dataset. Note that higher probability at a pixel denotes a higher success rate when it is attacked. We can observe patterns in $P_{\text{attack}}$ aligned to our gradient overlap on (b) toy model and (d) ResNet-18.

Table 1: Top-1 test accuracy (%) after performing various adversarial attacks on every pixel $\mathcal{G}(p = 1.0)$, its subset $\mathcal{G}(p = 0.3)$, and their differences (i.e., diff) on CIFAR-10 dataset. The toy model (see Appendix A) and ResNet-18 achieved 81.4% and 94.6% top-1 test accuracy, respectively. Note that all the diffs are close to zero.

| | Toy Model | | | ResNet-18 | | |
| Attack Methods | $p = 1.0$ | $p = 0.3$ | diff | $p = 1.0$ | $p = 0.3$ | diff |
|---|---|---|---|---|---|---|
| OnePixel (Su et al., 2017) | 56.6 | 58.4 | 1.7 | 57.2 | 59.5 | 2.4 |
| JSMA (Papernot et al., 2016a) | 0.2 | 0.4 | 0.2 | 3.2 | 9.8 | 6.6 |
| DeepFool (Moosavi-Dezfooli et al., 2016) | 18.5 | 18.6 | 0.1 | 7.2 | 11.5 | 4.3 |
| CW (Carlini & Wagner, 2016) | 0.0 | 0.0 | 0.0 | 0.0 | 0.0 | 0.0 |
| PGD (Madry et al., 2017) | 0.0 | 1.6 | 1.6 | 0.0 | 0.0 | 0.0 |

## 3.2 ADVERSARIAL ATTACK VIA GRADIENT OVERLAPS

To show that the pixels with high gradient overlaps are indeed a weak point of network, we generate adversarial examples on $\mathcal{G}(p)$. We evaluate top-1 accuracy of our toy model (the model defined as in the previous subsection) and ResNet-18 on CIFAR-10 dataset after performing five adversarial attacks (Su et al., 2017; Papernot et al., 2016a; Moosavi-Dezfooli et al., 2016; Carlini & Wagner, 2016; Madry et al., 2017) for $p \in \{1.0, 0.3\}$ (Table 1). Interestingly, constraining the domain of the attacks to $\mathcal{G}(0.3)$ barely decreases the success rate compared to the attacks on $\mathcal{G}(1.0)$.

We can observe that the pixels with the high gradient overlaps are more susceptible (i.e., likely to be in a vulnerable domain) to the adversarial attacks. Considering all the observations, we leverage the vulnerable domain of the pixels for adversarial defense. If we can intentionally impose the GCA onto a model input and let GCA occupy the vulnerable domain, we can fully induce the attacks on it so that the induced attacks can be dodged easily by a single padding operation.

## 4 ACE (ARTIFICIAL CHECKERBOARD ENHANCER)

### 4.1 MODULE DESCRIPTION

In this section, we propose the Artificial Checkerboard Enhancer (ACE) module, which artificially enhances the checkerboard pattern in the input gradients so that it induces the vulnerable domain to have the identical pattern. Figure 3a illustrates our proposed ACE module, which is based on a convolutional autoencoder. The encoder consists of convolutional layers where the first layer's $k$ is not divisible by $s$ ($k \not\equiv 0 \mod s$), for example, when $k = 1$ and $s = 2$. In order to preserve the information of the input $x$, we add an identity skip connection that bypasses the input of ACE module to the output. The hyperparameter $\lambda$ is to control the magnitude of checkerboard artifacts in the input gradients.

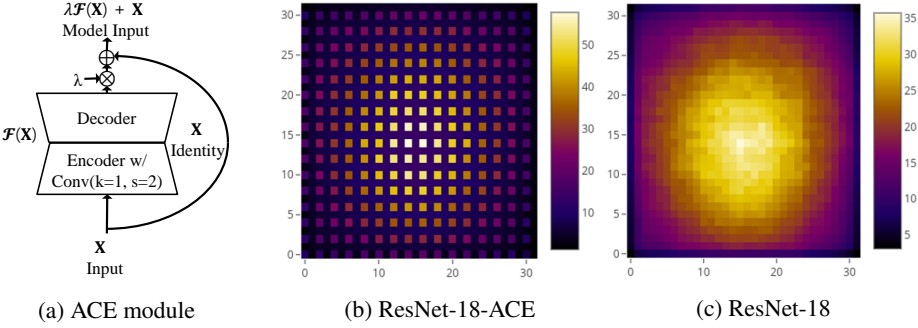

Figure 3: Our proposed ACE module and the gradient heatmaps of ResNet-18. (a) The schematic illustration of the ACE module. (b) ResNet-18-ACE ($\lambda = 10$) and (c) ResNet-18.

Table 2: Top-1 test accuracy (%) and the proportion (%) of the pixels having checkerboard artifacts ($\mathcal{C}$) in the top-30% gradient overlap $\mathcal{G}(0.3)$ (i.e., $|\mathcal{G}(0.3) \cap \mathcal{C}|/|\mathcal{C}|$) of the trained models on CIFAR-10 dataset with varying $\lambda$. Note that $\lambda = 0$ and $\lambda = \infty$ are equivalent to the vanilla model without the ACE module and the identity skip connection, respectively.

| | $\lambda$ | 0 | 1 | 2 | 5 | 10 | 20 | 100 | $\infty$ |
|---|---|---|---|---|---|---|---|---|---|
| Accuracy (%) | ResNet-18 + ACE | 94.6 | 94.4 | 94.3 | 94.4 | 94.5 | 92.4 | 92.0 | 91.0 |
| | VGG-11 + ACE | 91.8 | 91.7 | 92.3 | 91.5 | 89.8 | 88.8 | 88.5 | 87.1 |
| $\frac{|\mathcal{G}(0.3) \cap \mathcal{C}|}{|\mathcal{C}|}$ (%) | ResNet-18 + ACE | 31.3 | 34.4 | 35.9 | 44.9 | 53.9 | 63.3 | 86.7 | 100 |
| | VGG-11 + ACE | 5.1 | 63.3 | 64.5 | 66.8 | 70.7 | 75.4 | 93.8 | 100 |

We plug our ACE module in front of a base convolutional network to enhance the checkerboard artifacts on its input gradients. By increasing $\lambda$, we can artificially increase the gradient checkerboard artifacts of the network. Figure 3b and 3c show the heatmaps of the input gradients of ResNet-18 when $\lambda = 10$ and $\lambda = 0$ (i.e., without ACE module), respectively. The heatmap is generated by a channel-wise absolute sum of input gradients. Note that the checkerboard pattern is clearly observed in Figure 3b.

By changing the value of $\lambda$, we report the top-1 test accuracy and the proportion of the pixels having checkerboard artifacts ($\mathcal{C}$) in the top-30% gradient overlaps $\mathcal{G}(0.3)$ (Table 2). More precisely, we denote $\mathcal{C}$ as the GCA imposed by ACE module, which is identical to the set of pixels that are connected to its first convolutional layer of $k = 1$ and $s = 2$. In Table 2, we can observe that 1) there is only a small decrease in the accuracy even with a large $\lambda$ and 2) the pixels with the large gradient overlaps gradually coincide with the GCA as the $\lambda$ increases. Furthermore, according to the results in Table 1, the existing adversarial attacks tend to be induced on the pixels with the high gradient overlaps. Therefore, we can conjecture that our ACE module which builds a high gradient overlap with a significant checkerboard pattern could cage the adversarial attacks into the checkerboard artifacts, and this will be empirically proven in Section 5.

## 4.2 PARAMETER STUDY

We now study the effects of $\lambda$. To this end, we first visualize the classified labels with respect to the magnitude of the perturbation on pixels in checkerboard artifacts $\mathcal{C}$ and pixels in non-checkerboard artifacts $X \backslash \mathcal{C}$. Let $x$ be the input image and $\hat{\mathbf{e}}_{\mathcal{C}} = \frac{M_{\mathcal{C}} \odot \nabla x}{\|M_{\mathcal{C}} \odot \nabla x\|_F}$ where $M_{\mathcal{C}}$ denotes a mask (i.e., value of $i$-th element of $M_{\mathcal{C}}$ equals to one if $x_i \in \mathcal{C}$ and zero otherwise) and $\odot$ denotes the element-wise multiplication. We define $\hat{\mathbf{e}}_{X \backslash \mathcal{C}}$ in a similar way.

We plot the classified label map of $x + \bar{i}\hat{\mathbf{e}}_{X \backslash \mathcal{C}} + \bar{j}\hat{\mathbf{e}}_{\mathcal{C}}$ by varying $\bar{i}$ and $\bar{j}$. For the experiment, we first train ACE module as an autoencoder using ImageNet (Deng et al., 2009) datasets and plug ACE module into pretrained ResNet-152 as described in the following experiment section. Next, we plot

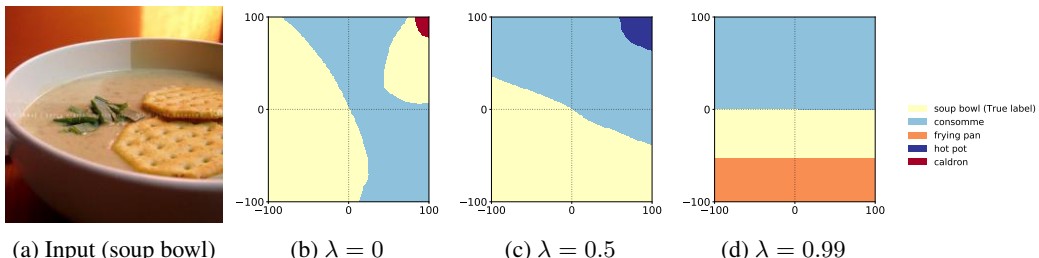

(a) Input (soup bowl)  (b) $\lambda = 0$  (c) $\lambda = 0.5$  (d) $\lambda = 0.99$

Figure 4: Classified label map after pixel perturbations. Using an example image in ImageNet dataset, we put (a) as the input image. The classified label maps corresponding to the different $\lambda$ are plotted in (b), (c), and (d), respectively. The same colors in (b), (c), and (d) denote the same classified labels. The center (i.e., the position of $(0, 0)$) denotes the label of the original image (a) without any perturbation. X-axis and Y-axis denote the perturbations of the pixels not in $\mathcal{C}$ and the pixels in $\mathcal{C}$, respectively.

classified labels for a sample image from ImageNet by varying $\bar{i}$ and $\bar{j}$ from $-100$ to $100$ by interval 1 (i.e., we test 40k perturbed images per each image) in Figure 4. The figure signifies the classified label map with respect to the perturbation on $X \backslash \mathcal{C}$ and $\mathcal{C}$. Without using our ACE module (when $\lambda = 0$), the perturbation through non-artifact pixels and artifact pixels similarly affect the label map. However, when $\lambda > 0$, we can observe that the artifact pixels are susceptible to change their labels with only a small perturbation while non-artifact pixels are robust to the same perturbation. Note that the asymmetry between the artifact pixels and non-artifacts becomes more clear as $\lambda$ increases.

### 4.3 Adversarial Defense with ACE

Here, we propose a novel defense method using ACE module. First, we plug ACE module into the input of a given network which enhances gradient checkerboard artifacts. Next, we let the adversary generate adversarial images using the network with ACE module. Because ACE module is likely to expose the pixels in the vulnerable domain, which is empirically shown in Appendix I, we may consider the ACE module as an inducer to the adversarial perturbations generated by the adversary into the checkerboard. Interestingly, because the pixels in the vulnerable domain are going to be aligned to the repeated checkerboard pattern, by shifting a single pixel of the adversarial sample, we can move perturbations into the non-vulnerable domain (i.e., non-artifact pixels). The proposed defense mechanism is similar to the defense method introduced in Xie et al. (2017). However, thanks to our ACE module, the vulnerable pixels are induced to checkerboard artifacts so that only **one-pixel padding** is enough to avoid several adversarial attacks aiming the pixels in the vulnerable domain. We also report the defense results regarding the diverse padding-sizes in Appendix D. It is worthwhile to note that large $\lambda$ induces significant gradient checkerboard artifacts hence leads to more robust defense. The detailed results are reported in Section 5.

## 5 Experimental Results

For thorough evaluation, we evaluate our proposed defense method in the following three attack scenarios, which are vanilla, transfer, and adaptive attacks. First, in the **vanilla attack** scenario, the adversary has access to the target model but not our proposed defense method (i.e., single pixel padding defense). We remark that the vanilla attack scenario is similar to the scenario used by Xie et al. (2017). Second, in the **transfer attack** scenario, the adversary generates adversarial perturbations from a source model, which is different from the target model. Finally, in the **adaptive attack** scenario, the adversary knows every aspect of the model and the defense method so that it can directly exploit our defense. For our experiments, Expectation Over Transformation (EOT) (Athalye et al., 2018) is used for the adaptive attack scenario. For the evaluation of our defense method, we use the following five attack methods as our adversary. OnePixel (Su et al., 2017), JSMA (Papernot et al., 2016a), DeepFool (Moosavi-Dezfooli et al., 2016), CW (Carlini & Wagner, 2016) and PGD (Madry et al., 2017)[1]. In addition, we conduct experiments on CIFAR-10 (Krizhevsky, 2009) and ImageNet (Deng et al., 2009) datasets for the attack scenarios.

For the models evaluated in CIFAR-10 dataset, we train the models with the two layered ACE module from scratch. Note that the first convolutional layer in ACE has $k = 1$ and $s = 2$, and the following deconvolutional layer has $k = 3$ and $s = 2$ so that it enhances gradient checkerboard artifacts. We would like to recall that the top-1 accuracy of VGG-11 (Simonyan & Zisserman, 2015) and ResNet-18 (He et al., 2016) with ACE module in respect of different $\lambda$ are reported in Table 2. Meanwhile, for a large-scale dataset such as ImageNet, training an entire network is very expensive. Hence, we train the ACE module as autoencoder with UNet architecture (Ronneberger et al., 2015) and plug ACE into the input of a pretrained network without any additional training procedure. In order to retain the scale of the input image of the pretrained network, we slightly modify the ACE module by constraining $\lambda \in [0, 1]$ and multiplying $(1 - \lambda)$ to the identity skip connection. In this way, our ACE module becomes capable of handling large-scale datasets in a plug and play fashion on any model.

We now introduce two evaluation metrics named **attack survival rate** and **defense success rate** which denote that top-1 accuracy after attack divided by the original top-1 accuracy of the model

---

[1]We set the number of iterations of PGD ($l_2$) attacks as $1,000$ by following the official code (https://github.com/anishathalye/obfuscated-gradients) released by the authors (Athalye et al., 2018).

Table 3: Attack survival rate (%) and defense success rate (%) (larger is better) by one-pixel padding defense on CIFAR-10 dataset with varying $\lambda$. Note that $\lambda = 0$ is the equivalent setting to single padding pixel experiments in Xie et al. (2017).

| Dataset | Model | | Attack survival rate | | | Defense success rate | | |
|---------|-------|------|------|------|------|------|------|------|
| | | $\lambda$ | 0 | 10 | 100 | 0 | 10 | 100 |
| CIFAR-10 | ResNet-18 | OnePixel | 68.3 | 62.2 | 47.9 | 80.0 | 92.9 | **99.7** |
| | | JSMA | 1.9 | 1.2 | 0.0 | 71.9 | 97.4 | **99.5** |
| | | DeepFool | 8.8 | 8.8 | 11.7 | 82.6 | 99.1 | **99.7** |
| | | CW | 0.0 | 0.0 | 0.0 | 85.4 | 99.0 | **99.2** |
| | | PGD | 0.0 | 0.0 | 0.0 | 0.0 | 32.1 | **98.3** |
| CIFAR-10 | VGG-11 | OnePixel | 50.2 | 23.8 | 22.5 | 71.5 | 98.2 | **99.4** |
| | | JSMA | 0.0 | 0.0 | 0.0 | 61.3 | 99.1 | **99.8** |
| | | DeepFool | 8.8 | 11.1 | 13.7 | 80.5 | 99.3 | **99.8** |
| | | CW | 0.0 | 0.0 | 0.0 | 95.9 | 99.2 | **99.7** |
| | | PGD | 0.0 | 0.0 | 0.0 | 0.1 | 85.2 | **98.8** |

Table 4: Attack survival rate (%) and defense success rate (%) (larger is better) by one-pixel padding defense on ImageNet dataset with varying $\lambda$.

| Dataset | Model | | Attack survival rate | | | Defense success rate | | |
|---------|-------|------|------|------|------|------|------|------|
| | | $\lambda$ | 0 | 0.5 | 0.99 | 0 | 0.5 | 0.99 |
| ImageNet | ResNet-152 | CW | 0.0 | 0.0 | 0.0 | 69.3 | 86.1 | **88.9** |
| | | PGD | 0.0 | 0.0 | 0.0 | 0.0 | 3.2 | **88.1** |
| ImageNet | VGG-19 | CW | 0.0 | 0.0 | 0.0 | 40.3 | 70.2 | **85.1** |
| | | PGD | 0.1 | 0.0 | 0.0 | 0.1 | 0.0 | **84.3** |

Table 5: Defense success rate (%) for varying $\lambda$ after the PGD attack on CIFAR-10 dataset.

| $\lambda$ | 0 | 1 | 2 | 5 | 10 | 20 | 100 |
|-----------|------|------|------|------|------|------|------|
| ResNet-18-ACE | 0.0 | 0.0 | 10.2 | 23.9 | 32.1 | 93.5 | **98.3** |
| VGG-11-ACE | 0.8 | 1.7 | 1.9 | 9.7 | 85.9 | 79.5 | **98.8** |

and top-1 accuracy after defending attack divided by the original accuracy, respectively. We note that all experimental results reported in this section are reproduced by ourselves[2].

**Vanilla attack scenario.** We evaluate our defense method in both CIFAR-10 and ImageNet dataset. For CIFAR-10 experiments, we train an entire network including ACE. For ImageNet experiments, we only train ACE module as conventional autoencoder by minimizing the mean squared error without training an entire network. Table 3 shows that our proposed method defends various attack methods successfully on CIFAR-10 dataset. We remark that by choosing a large $\lambda$ (e.g., $\lambda = 100$), the top-1 accuracy after defense on performed attacks is barely dropped. In table 4, we report same experiments conducted on ImageNet dataset. We use the pretrained models of VGG-19 and ResNet-152[3] repository whose top-1 test accuracy on ImageNet dataset is 72.38% and 78.31%, respectively. We abbreviate the results of OnePixel, JSMA and DeepFool due to the infeasibly high cost of time and memory limit for those algorithms to craft numerous adversarial images in ImageNet dataset. Comparison with other defense methods (Prakash et al., 2018; Xie et al., 2017; Cao & Gong, 2017) for CIFAR-10 and ImageNet datasets are reported in Appendix E. To investigate the effectiveness of $\lambda$ regards to defense success rate, we evaluate PGD attack for $\lambda \in \{0, 1, 2, 5, 10, 20, 100\}$ and report the defense success rates in Table 5. The result shows that

---

[2]We will publish the code at the final round of the paper.

[3]We used pretrained models from the official PyTorch repository (https://github.com/pytorch/vision).

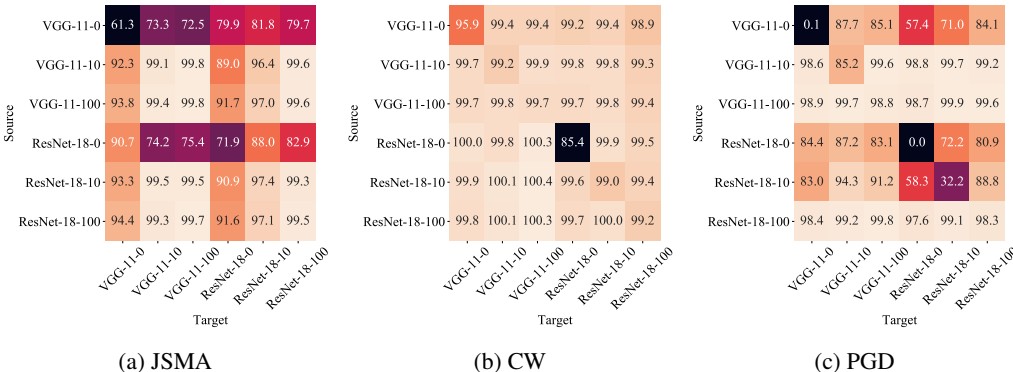

(a) JSMA          (b) CW          (c) PGD

Figure 5: Defense success rate (%) on CIFAR-10 test dataset after one-pixel padding defense on transfer attacks from the source model to the target model via JSMA, CW and PGD. The number followed by name of network denotes the intensity of $\lambda$, e.g., VGG-11-100 denotes VGG-11 + ACE with $\lambda = 100$. Note that adversarial examples generated by different $\lambda$ are not transferable to other models.

when $\lambda$ increases, the defense success rate improves as well. To the best of our knowledge, this is the first work that defends PGD up to $98\%$.

**Transfer attack scenario.** It has been demonstrated that conventional deep networks are vulnerable to transfer attacks proposed by Papernot et al. (2016b). To show that our method is robust to transfer attacks, we conduct transfer attacks for VGG-11 and ResNet-18 by choosing $\lambda \in \{0, 10, 100\}$ on CIFAR-10 dataset. We report the defense success rate after one-pixel padding defense of transfer attacks by JSMA, CW and PGD in Figure 5. More results including OnePixel and DeepFool transfer attack experiments are reported in Appendix G. The results show that generated adversarial samples are not transferable to other models with different $\lambda$.

**Adaptive attack scenario.** A successful defense method should defend $l_0$, $l_2$, and even $l_\infty$ bounded adversaries and also show robust results on the adaptive white-box setting. We report our defense results combined with adversarial training against Expectation Over Transformation (EOT) (Athalye et al., 2018) of PGD attack in Appendix H. From the results, it turns out that our method is complementary with robust training defense methods (e.g., adversarial training). Therefore, if we combine our method with the existing robust training defense methods together, we can secure promising results on the vanilla scenario and even perform well on the adaptive scenario.

# 6 CONCLUSION

In this paper, we have investigated the gradient checkerboard artifacts (GCA) which turned out to be a potential threat to the network robustness. Based on our observations, we proposed a novel Artificial Checkerboard Enhancer (ACE) which attracts the attack to the pre-specified domain. ACE is a module that can be plugged in front of any pretrained model with a negligible performance drop. Exploiting its favorable characteristics, we provide a simple yet effective defense method that is even scalable to a large-scale dataset such as ImageNet. Our extensive experiments show that the proposed method can deflect various attack scenarios with remarkable defense rates compared with several existing adversarial defense methods.

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

## A   NETWORK ARCHITECTURE OF OUR TOY MODEL

We present our toy model used in Section 3. The model consists of three convolutional layers followed by ReLUs and two fully-connected layers (Table 6).

Table 6: Architecture detail of our toy model used on CIFAR-10 dataset.

| Input image $x \in \mathbb{R}^{32 \times 32 \times 3}$ |
| :---: |
| $3 \times 3$, stride=2 conv16 ReLU |
| $3 \times 3$, stride=2 conv32 ReLU |
| $3 \times 3$, stride=1 conv64 ReLU |
| dense $1600 \rightarrow 512$ |
| dense $512 \rightarrow 10$ |

## B   CHECKERBOARD ARTIFACTS IN GRADIENTS

To visualize the existence of checkerboard artifacts in the gradients, here we present the average heatmaps over the test images. Let $\nabla x_{i,j,k}^l$ denote the feature gradient in layer $l$ where $i, j$ and $k$ are the indices of xy-axis and the channel, respectively. In each layer, the heatmap $\boldsymbol{h}^l$ is gathered by a channel-wise absolute sum of the feature gradients. More specifically, $h_{i,j}^l = \sum_{\text{dataset}} \sum_k |\nabla x_{i,j,k}^l|$, where $x^0$ is an input. Figure 6 and 7 show the feature gradients of our toy model and ResNet-18.



Figure 6: Gradient heatmaps of our toy model on CIFAR-10 dataset. Each heatmap shows the intermediate feature gradients. For example, conv1 denotes the feature after the first convolutional layer. We can observe the checkerboard patterns before the strided convolutional layers (input and conv1).

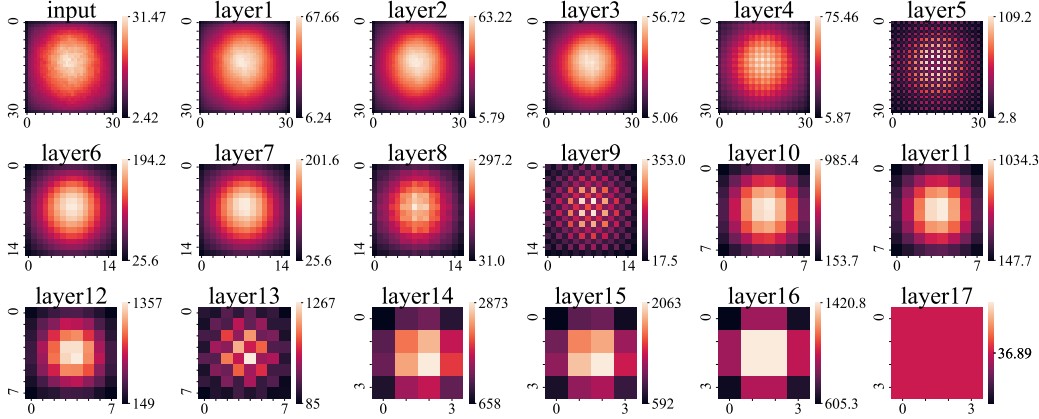

Figure 7: Aggregated feature gradient heatmaps of ResNet-18 on CIFAR-10 dataset. Each heatmap is the intermediate feature gradient in sequence. e.g., layer1 denotes feature gradient after 1st layer. We can observe checkerboard patterns before strided convolutional layers (layer 5, 9, 13) and they propagate to their front layers (layer 4, 8).

## C    GRADIENT OVERLAPS COMPUTATION

To count the gradient overlaps per pixel ($\Omega(x_i)$), for simplicity, we only consider convolutional and fully-connected layers. We set each and every network parameter to 1 to evenly distribute the gradients from the loss layer to the previous layers. Therefore, the gradients of the modified network are aggregated in a certain location (or pixel) if and only if there is a linear overlap from the later layers at the backward pass.

## D    STUDY ON THE PADDING-SIZE OF THE PROPOSED DEFENSE WITH ACE

In Table 7, we report the defense success rate varying the padding-size from one to five. We can observe that the proposed defense mechanism almost prevents accuracy drop with the padding-sizes of one, three and five (odd numbers). The result supports that ACE induces adversarial perturbations into checkerboard artifacts, which could be avoided by padding an image with one pixel.

Table 7: Defense success rate (%) for varying the padding-size on ACE architectures. $\lambda = 10$. Note that the padding-size of two or four significantly degrades the defense rate of the ACE models for various attack methods.

| Model | Attack Method | Padding-size | | | | |
|---|---|---|---|---|---|---|
| | | 1 | 2 | 3 | 4 | 5 |
| | OnePixel | **92.9** | 74.4 | 92.0 | 66.0 | 89.8 |
| | JSMA | **97.5** | 58.8 | 96.7 | 42.3 | 94.6 |
| ResNet-18 | DeepFool | **99.1** | 79.2 | 97.2 | 63.2 | 93.6 |
| | CW | **99.0** | 80.7 | 97.6 | 61.6 | 94.6 |
| | PGD | 32.1 | 0.0 | **35.9** | 0.0 | 35.8 |
| | OnePixel | **98.2** | 48.0 | 97.4 | 42.4 | 95.2 |
| | JSMA | **99.1** | 56.1 | 98.2 | 45.8 | 96.2 |
| VGG-11 | DeepFool | **99.3** | 87.0 | 98.5 | 68.5 | 96.0 |
| | CW | **99.2** | 89.2 | 98.4 | 69.4 | 96.0 |
| | PGD | **85.2** | 0.0 | 84.7 | 0.0 | 80.0 |

## E    COMPARISON WITH OTHER DEFENSE METHODS

We implement all the defense methods [4] (Prakash et al., 2018; Xie et al., 2017; Cao & Gong, 2017) by following their papers. For fair comparison, we follow the suggested settings in their papers, and the results are presented in Table 8 and Table 9. Specifically, for Prakash et al. (2018), with R-CAM implemented, the number of deflection is set to 100 with window-size of 10 and sigma for denoiser of 0.04, respectively. For Xie et al. (2017), the scale factor is set to 0.9. Finally, for Cao & Gong (2017), the number of the ensemble is set to 1000 and the radius of the region is set to 0.02.

## F    PARAMETER STUDY (CONT')

In this section, we plot the classified label map after pixel perturbation on artifact pixels and non-artifact pixels as described in Section 4.2. We follow the same setting of Section 4.2. The results are reported in Figure 8.

---

[4]We will publish the code at the final round of the paper.

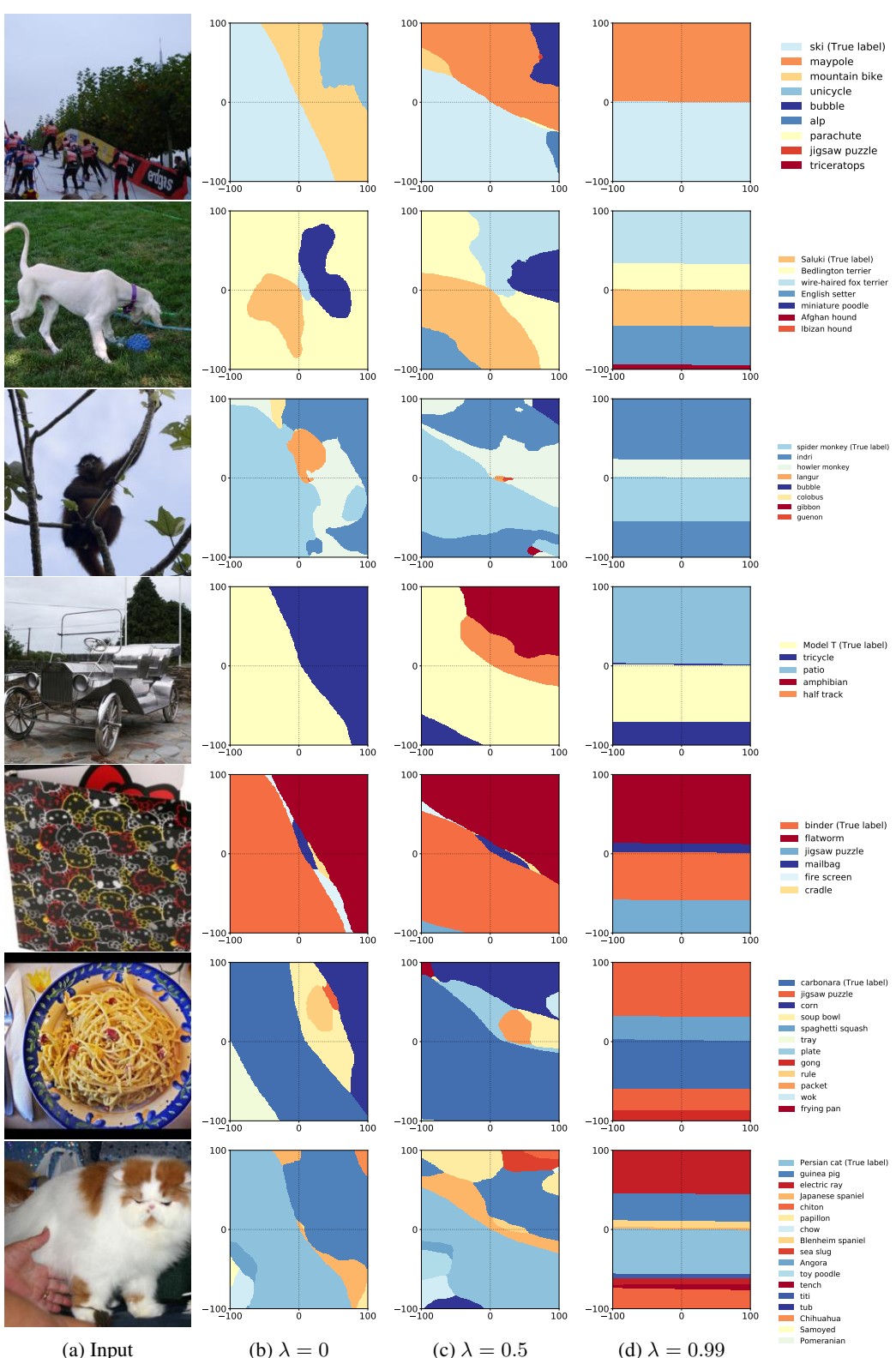

Figure 8: Classified label map after pixel perturbations described in Section 4.2. Each row is composed of an input image and the label maps for $\lambda$ equals $0$, $0.5$ and $0.99$.

Table 8: Top-1 accuracy (%) of four defense methods against five attack methods on CIFAR-10 dataset. Boldface denotes the highest top-1 accuracy after defense. $\lambda = 100$ for our method, one-pixel padding with the ACE module. ResNet-18 is used for the experiments.

| Attack Methods | Pixel Deflection (Prakash et al.) | Randomization (Xie et al.) | Region-Based (Cao & Gong) | Our Method |
|---|---|---|---|---|
| OnePixel | 60.7 | 76.9 | 65.1 | **91.7** |
| JSMA | 48.7 | 72.6 | 6.3 | **91.5** |
| DeepFool | 71.9 | 84.4 | 78.0 | **91.7** |
| CW | 66.7 | 74.7 | 7.6 | **91.3** |
| PGD | 10.6 | 1.5 | 0.0 | **90.4** |

Table 9: Top-1 accuracy (%) of four defense methods against two attack methods on 1,000 images in ImageNet dataset. Boldface denotes the highest top-1 accuracy after defense. $\lambda = 0.99$ for our method, single pixel padding with the ACE module. ResNet-152 is used for the experiments.

| Attack Methods | Pixel Deflection (Prakash et al.) | Randomization (Xie et al.) | Region-Based (Cao & Gong) | Our Method |
|---|---|---|---|---|
| CW | 67.4 | 69.5 | 0.0 | **69.6** |
| PGD | 23.8 | 14.1 | 0.1 | **69.0** |

## G  TRANSFER ATTACK RESULTS

Here, we present the results for defense success rate (%) on transfer attacks using VGG-11, ResNet-18 with various $\lambda$. Recall from Section 5, the result shows that generated adversarial samples are not effectively transferable on various $\lambda$. Source/target model with $\lambda$ is labeled in the format of [model]-[$\lambda$] (e.g., VGG-11-100 denotes VGG-11 with ACE $\lambda = 100$).

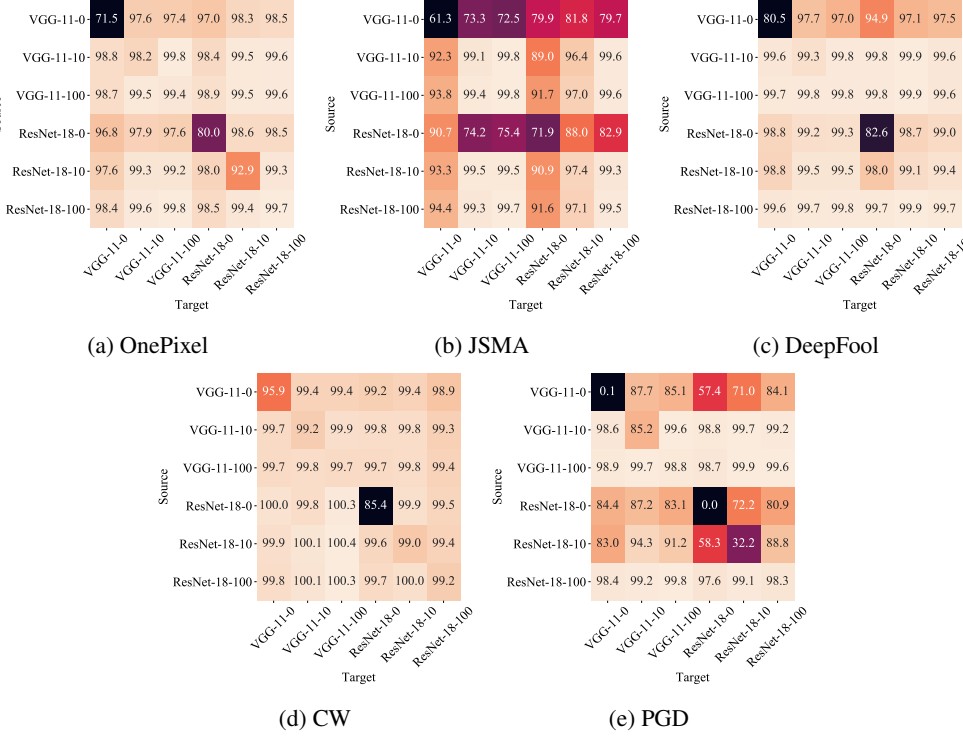

Figure 9: Defense success rate (%) on the transfer attacks with five adversaries.

## H    DEFENSE RESULTS FOR THE ADAPTIVE CASE

**Adaptive attack case**    A successful defense method should be able to defend various conditions including $l_0$, $l_2$ and $l_\infty$-bounded adversaries as well as an adaptive white-box setting where the adversary knows our defense method in every aspect. Under the adaptive white-box setting, we conducted experiments in Table 10. In order to avoid direct exploitation of our padding direction, we shift our images in the random direction around the known safe points near our checkerboard artifacts. By combining PGD adversarial training (Madry et al., 2017) for robustness on $l_\infty$ bounded attacks to our method, we can defend the corresponding adaptive attack for stochastic methods known as Expectation Over Transformation (EOT, Athalye & Sutskever (2017)). This method was used to break Xie et al. (2017) in Athalye et al. (2018). Although we have some loss in Top-1 accuracy when $\lambda$ is high, we have advantages in that we can defend vanilla attack cases at the same time.

Table 10: Top-1 accuracy (%) after EOT attack on our defense method together with adversarial training on CIFAR-10 dataset. All images were padded by one-pixel to X-axis. Conducted attacks are written in the format of PGD-norm-iterations. ACE module only shows training accuracy loss due to high $\lambda$.

| Dataset | Model | | A (Target model) | | |
|---------|-------|--------------|------|------|------|
| | | $\lambda$ | 0 | 10 | 100 |
| CIFAR-10 | ResNet-18 | PGD-$l_\infty$-10 | 83.0 | 83.0 | 79.6 |
| | | PGD-$l_\infty$-100 | 68.1 | 67.8 | 63.1 |
| | | PGD-$l_\infty$-1000 | 50.7 | 50.0 | 45.1 |
| | | PGD-$l_2$-10 | 80.2 | 79.6 | 78.5 |
| | | PGD-$l_2$-100 | 57.9 | 57.1 | 56.8 |
| | | PGD-$l_2$-1000 | 48.6 | 47.8 | 43.7 |
| CIFAR-10 | VGG-11 | PGD-$l_\infty$-10 | 81.7 | 78.1 | 78.1 |
| | | PGD-$l_\infty$-100 | 67.0 | 62.1 | 62.8 |
| | | PGD-$l_\infty$-1000 | 44.4 | 41.3 | 42.8 |
| | | PGD-$l_2$-10 | 77.7 | 76.7 | 76.4 |
| | | PGD-$l_2$-100 | 50.3 | 53.7 | 53.5 |
| | | PGD-$l_2$-1000 | 41.4 | 39.1 | 40.1 |

## I    PERTURBATION MAP ON VARYING ARTIFACT INTENSITY

Let $x$ and $x_{avd}$ be the pure input image and the adversarial sample generated by an adversary, respectively. Figure 10 and 11 depict the average difference between $x_{avd}$ and $x$ over CIFAR-10 for ResNet-18 and VGG-11. From the figures, we can clearly observe significant checkerboard artifacts when enlarging $\lambda$. The observation supports our conjecture that most of the adversarial perturbations show checkerboard patterns when ACE is plugged into the models.

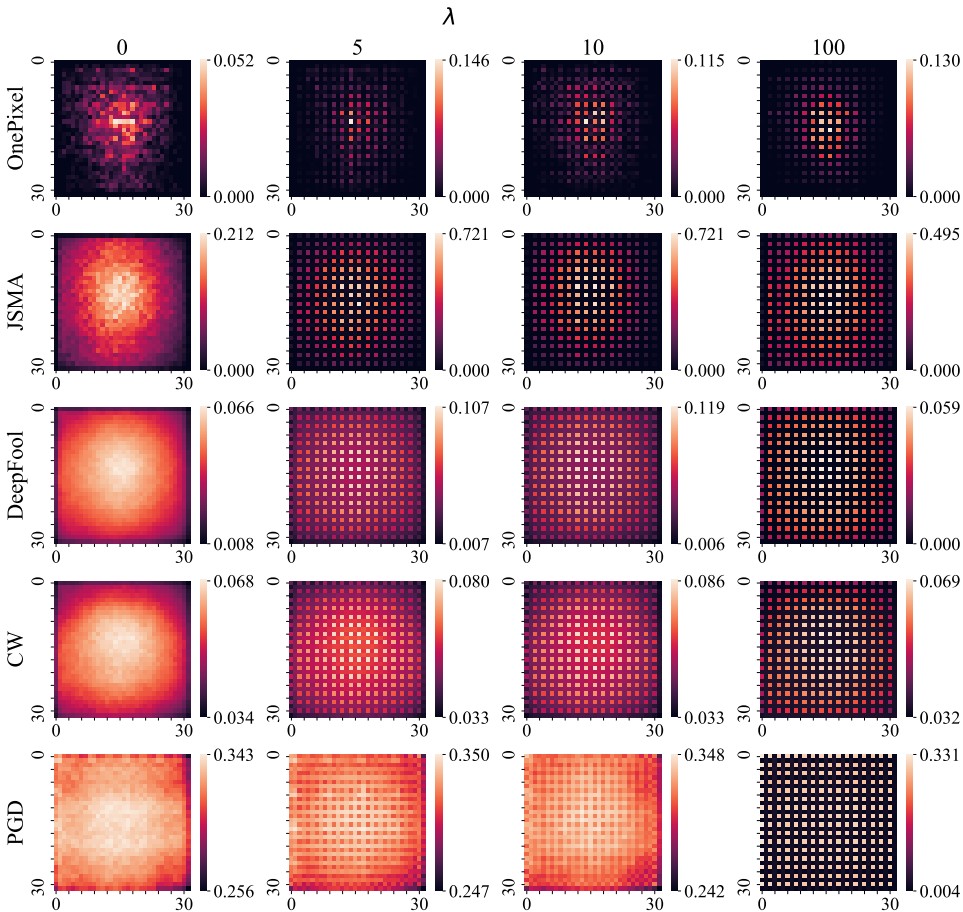

Figure 10: Aggregated perturbation heatmaps of ResNet-18 with varying $\lambda$ on CIFAR-10 dataset.

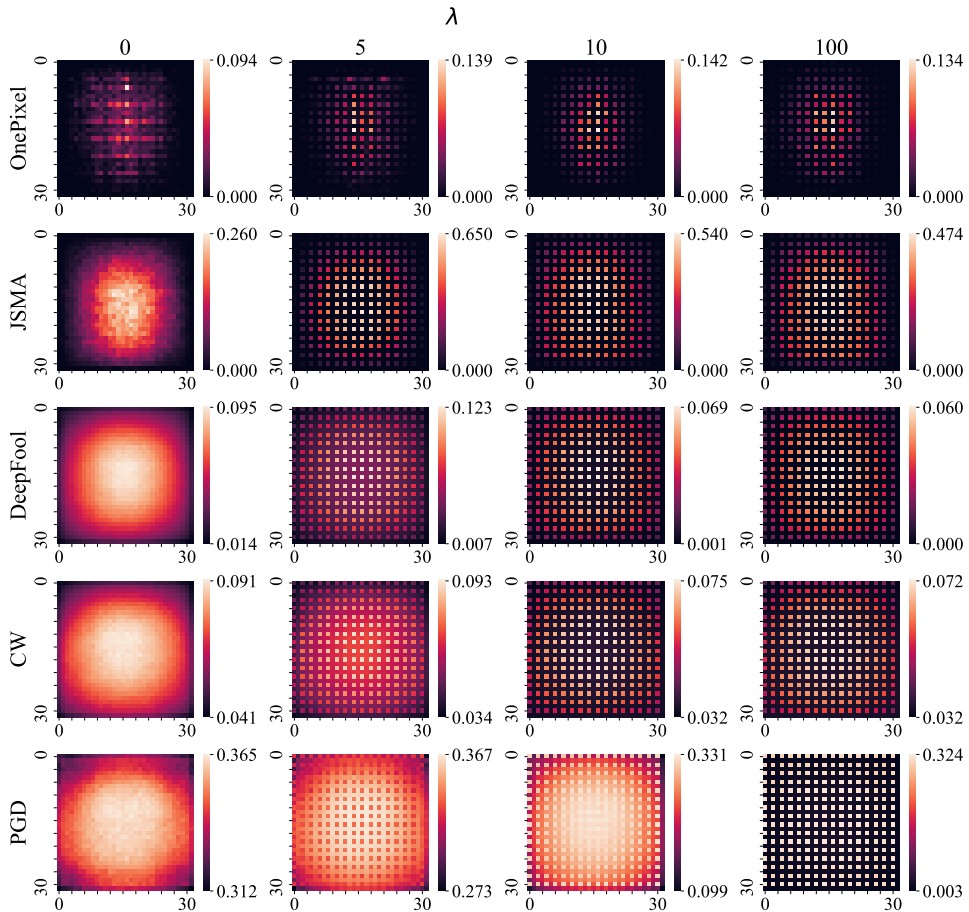

Figure 11: Aggregated perturbation heatmaps of VGG-11 with varying $\lambda$ on CIFAR-10 dataset.

