# OpenReview forum: "ACE: Artificial Checkerboard Enhancer to Induce and Evade Adversarial Attacks"
_ICLR.cc/2019/Conference_

### Official Review · AnonReviewer2 · 2018-10-22
**I am not right person to review**

**Rating:** 6
**Confidence:** 1

**Review:**

I am a researcher in NLP and know little about vision, so I cannot review this paper. I have contacted general chair about this situation.

---

### Official Review · AnonReviewer3 · 2018-10-30
**I worry that this defense will be easy to break**

**Rating:** 4
**Confidence:** 3

**Review:**

The authors propose that the "checkerboard phenomenon", whereby the gradients exhibit a repeating pattern over the pixel space, is a source of vulnerability to adversarial examples. They propose to first enhance this vulnerability for pre-trained models with a pre-conditioning layer, and then to evade it by zero padding the image to offset the pattern.

Clarity: I found the work difficult to follow in places, and I felt that some material crucial to the paper was relegated to appendices.

Originality: To my knowledge, the idea is original.

Quality and significance:
I feel the significance of this work is likely to be low. While the authors report positive "defense" results, I strongly suspect this is simply because the attacks considered did not uncover the defense strategy. I expect that this defense would be broken relatively quickly if the paper is accepted. The authors did not present evidence to suggest that their method reduces the probability that a misclassified example lies close to the training or test examples. As such, the defense seems to rely on the attacker being "tricked".

Specific comments:

1) Throughout the paper, I was unclear what specific kinds of attacks the method was intended to defend against.
2) In section 2.2, key definitions are relegated to appendix C.
3) Section 3.2-> p = 0.3 is still 30% of the pixels. Could the authors provide a baseline with a random 30% of pixels?
4) Adaptive attack scenario: I would recommend that the authors also included a measure of the attack-ability under random noise in the epsilon ball. This would demonstrate whether the defense actually removes adversarial examples or just "attacks the attackers".

---

> ### Author Response · Authors · 2018-11-08
> **Our response to reviewer3**
>
> We thank the detailed review and valuable comments. First of all, we believe that our defense will not be easy to break even if the adversary knows in advance about our method. We hope that this response can resolve all of your concerns. Plus, we will revise the overall paper organization for better clarity.
>
> Before responding to the comments (from C1 to C6 below), we first point out the relationship between building a “robust model” and creating a “successful defense method”. The Euclidean distance between the fooled images and the originals (in C2 below), and the attack-ability inside an epsilon ball (in C6 below) are good measures for model robustness in metric spaces with the l_2-norm and its equivalent. The attack methods that are formulated within the l_2-equivalent norms, such as Carlini&Wagner and PGD, can be defended if a model is robust concerning the measures above. However, for the attacks including OnePixel and JSMA, which are not formulated within the l_2-equivalent, the model can be fooled even when the model is robust concerning those measures. This is due to the fact that the constraint these attacks utilize cannot be bounded by the l_2-norm. Therefore, a “robust model” is robust to l_2-equivalent attacks and can be considered as a subset of “successful defense methods”. This can be demonstrated by evaluating the classification accuracy of the models adversarially-trained with PGD, when attacking with an l_0-norm-based attack.
>
>
> C1) The attacks considered did not uncover the defense strategy.
> -> At the beginning of Section 5, we propose three types of threat models for attack algorithms. Among those attack scenarios, we considered the “white-box scenario” where the attacks uncover our defense strategy. To show that our method is not vulnerable in the scenario, we conducted the experiment against the PGD attack method, which is the strongest attack, combined with adversarial training. As shown in Appendix H, our method can successfully defend the attacks, thus we can guarantee our method can successfully defend in the white-box scenario as well.
>
> C2) Evidence to suggest that the method can reduce the probability that a misclassified example lies close to the training or test examples was not presented.
> -> Using the suggested measure concerning the probability change could be suitable to identify the l_2-equivalent-robustness of a model. However, as mentioned above, since the probability cannot reflect the robustness against attacks that are not formulated within the l_2-equivalent such as OnePixel and JSMA attacks, our method did not aim to reduce this probability.
>
> C3) The specific kinds of attacks the method was intended to defend against were unclear.
> -> Our goal is to propose a method that can be utilized in both black-box and white-box attack scenarios as explicitly mentioned in Section 5. Specifically, we targeted three scenarios: 1) the vanilla attack scenario, the adversary can access the target model but not our proposed defense method, 2) the transfer attack scenario, the adversary generates adversarial perturbations from a source model, which is different from the target model, and finally 3) the adaptive attack scenario (white-box attack scenario), the adversary knows every aspect of the model and the defense method so that it can directly exploit our defense.
>
> C4)  In section 2.2, key definitions are relegated to Appendix C
> -> We will revise our paper for better readability as you suggested.
>
> C5) Could the authors provide a baseline with a random 30% of pixels?
> -> Yes, we have been conducting an experiment based on the suggested setting. We will post the result as soon as possible.
>
> C6) Recommendation about including a measure of the attack-ability under random noise in the epsilon ball.
> -> Thank you for the suggestion. It seems that we need to verify the robustness of our method against the attacks that do not utilize the gradients. Our method would be robust against such attacks because of the following three reasons. First, for a random noise in the epsilon ball, the probability that the noise can directly affect the labels is very low (according to Section 4 in [1]). Furthermore, the probability that it matches the “random” direction of shift (i.e., in the adaptive scenario) as mentioned in Appendix H is low as well. Finally, the random noise itself can be reduced through the autoencoder structure of the ACE module (according to Section 5 in [2]). We will conduct experiments by following your suggestion to verify these claims and strengthen our proposed defense method.
>
>
> [1] Xiaoyu Chao and Neil Zhenqiang Gong. “Mitigating Evasion Attacks to Deep Neural Networks via Region-based Classification”. ACSAC 2017.
>
> [2] Pascal Vincent, Hugo Larochelle, Yoshua Bengio, and Pierre-Antoine Manzagol. “Extracting and Composing Robust Features with Denoising Autoencoders”. ICML 2008.

---

> > ### Comment · AnonReviewer3 · 2018-11-27
> > **Rating unchanged**
> >
> > I thank the authors for their detailed response. Unfortunately my assessment remains unchanged:
> >
> > Regarding robustness versus successful defense. Any attack should obey a set of constraints (which define the sense in which the attack is adversarial). This could be a bounded norm, or the set of rotations of the image, or a number of pixels that can be changed, etc. A meaningful defense should be robust, in that all points which obey the constraints are correctly classified. If the defense is not robust, then its success represents the limitations of the attacks used. (one counter-case would be if you could prove that the remaining adversarial points inside the constraints are computationally hard to find).
> >
> > I would change my assessment, if the authors provided convincing evidence that the defense is robust. I acknowledge that the authors considered an attacker which is aware of the defense, but it is not clear to me that this attacker successfully exploited this information.

---

> > > ### Author Response · Authors · 2018-12-08
> > > **Thank you for the suggestion**
> > >
> > > Finding the constraint on which our model is robust is crucial. Thank you for the suggestion.
> > >
> > > For now, from our experiments we find that our model is robust against  L0-based attacks. Our method works well for attacks that are not bounded within an epsilon ball, but are bounded in terms of the number of pixels perturbed.
> > >
> > > We will come up with a more general measure of robustness to backup the performance of our model.

---

### Official Review · AnonReviewer4 · 2018-11-17
**Needs further clarifications**

**Rating:** 4
**Confidence:** 2

**Review:**

I have to emphasize first that this is not my area of expertise so I am going to review it as an outsider.

The authors argue that the checkerboard phenomenon can be exploited to make neural networks robust against adversarial attacks. They propose to enhance the checkerboard pattern by first adding a layer, called Artificial Checkerboard Enhancer (ACE), and then evading the attacks by zero-padding the image. The authors’ argument is that enhancing the checkerboard phenomenon will make attacks more targeted towards certain pixels, which can be evaded by shifting the image.

Overall, I think the paper is difficult to read and is not suitable for publication. In terms of clarity, the authors do not use precise terminology that would allow the reader to reproduce their work. They allude to vague statements. For example, they introduce two KEY terminologies that are repeatedly used throughout the paper but are not properly defined (see for instance the “definition” of “Gradient Overlap” in Appendix C).

In addition, in terms of the experiements, it certainly does not help to say that they were “reproduced by [the authors themselves]”. What does this mean?

In terms of originality, I agree with the first reviewer that the defense strategy seems to be easily breakable. The authors propose that they enhance the checkerboard phenomenon so that adversarial attacks become easier to implement by targeting individual pixels (the pixels in the checkerboard artifacts). Then, they pad the image with zero pixels to shift it to the right. I don’t understand how shifting the pixels would make it harder to attack (especially when the adversary knows the system).

It would be really appreciated if the authors elaborate on the following points to help me understand their contribution:
- The entire discussion about ACE in Section 4.1 is ad-hoc and not well-motivated. Why would ACE enhance the checkerboard patterm? Can you please explain why it works? This is not mentioned anywhere in the paper. The experiment in Section 4.2 helps a bit but it does not answer this question.

- What wouldn't an adversary remove the padded pixels before generating the attack? In defense strategies, it is often assumed that the adversary knows the system. Can you please explain why that is not possible in this setting?

-

In Figure 4, the axes are \bar i and \bar j in the main body, but they are x and y in the figure. Please use the same notation.

---

> ### Author Response · Authors · 2018-11-23
> **Our response to reviewer4**
>
> Thank you for reviewing our paper with valuable comments. We will revise the terminology to be more precise in the paper as you suggested. We will present the definition of key ideas as a formula, and the specifications of experiments will be documented in our code for the ease of reproduction.
>
> C1) The meaning of “they were reproduced by [the authors themselves]”
> -> We intended to show clearly that all the experiments are reproduced by ourselves. This is just to stress that our results are reproducible and to present that we will release our code so that everyone can easily reproduce our results.
>
> C2) How shifting the pixel makes it harder to attack in the adaptive case.
> -> In Appendix H, the ACE module “randomly” shifts the pixel so that the adversary knows the distribution of shift, but does not know the exact direction for a single image. For attack algorithms bounded by l_0, l_1, or l_2-norm (i.e., except for the l_inf-norm-based attacks), this random shift averages out the perturbation within the neighborhood of each pixel. Then, the intensity of perturbation towards the decision boundary is reduced, therefore we have the increased probability of classifying the attacked image correctly, which is identical to increasing the defense rate.
>
> However in the l_inf-norm case, random shift does not necessarily reduce the intensity of perturbation imposed on each pixel. In this case, shifting the pixel may not have any influence on defense, thus the attack should be mitigated by the use of adversarial training. Therefore, we performed the adversarial training by combining our pixel shifting method as shown in Appendix H..
>
> C3) Why would ACE enhance the checkerboard pattern?
> -> This is because the ACE module makes our model learn through the checkerboard pattern with respect to our intensity parameter lambda. Let us assume that the ACE module has the autoencoder structure stated in the second paragraph in Section 5. If λ=0, the gradient clearly has no checkerboard pattern as shown in Figure 3(c). If λ=1, the gradient must be distributed in a checkerboard pattern as Figure 3(b). This is because these pixels are the only pixels that are connected to the output. For λ ∈ (0,1), the gradient can be interpreted as the interpolation between the case of λ=0 and λ=1. As λ approaches 1 from 0, the checkerboard pattern becomes more clear. Since the original network without the ACE module is the same as when λ=0 with the ACE module, using the ACE module with λ > 0 will enhance the checkerboard pattern in gradient. Please refer to Section 4.1 for details about the structure of ACE.
>
> C4) Why wouldn't an adversary remove the padded pixels before generating the attack?
> -> Yes it would. “The adversary in the adaptive case” in Appendix H does try to remove the padded pixels. We have set the direction of shift to random in order to avoid this.

---

### Public Comment · (anonymous) · 2018-10-24
**Figure 8 question**

Can you explain Figure 8? How are the X and Y axis selected?

It is confusing that traveling +/- 100 along the X axis does not change the class label, but traveling +/- along the Y axis quickly does.

---

> ### Author Response · Authors · 2018-10-27
> **Explanation on Figure 8**
>
> Thank you for your interest in our work. How Figure 8 was generated has been explained in Section 4.2, but we would like to elaborate more on this. We will explain about the axes in the figure first, and then give some detailed explanation on why the classified label map has such shape.
>
> For a classified label map (Figure 4 and 8), we have an input image x without any perturbation at (0, 0), where X-axis is the gradient direction vector of checkerboard artifacts (C), and Y-axis is the gradient direction vector of non-checkerboard artifacts (X\C). Formally, this is expressed as \hat{e}_C and \hat{e}_{X\C} in Section 4.2. Note that C is the checkerboard pixels with high absolute gradients designed by our ACE module with 1 x1 conv and stride 2 (Figure 3.(b) shows the pixels in C that absolute gradients turn out to be in general greater than those in X\C). Then, each point in the classified label map is generated by perturbing the original image to the direction of (x, y) coordinates from -100 to 100 respectively. We have the classified label map after pixel perturbations using an example image of soup bowl as shown in Figure 4.
>
> In Figure 4, we have empirically demonstrated the effect of this imbalance on gradients by creating a classified label map. This is an empirical backup of showing that our ACE module induces the vulnerable domain to the checkerboard. As \lambda increases, our model becomes more vulnerable on the perturbation on C, while the opposite behavior is observed on the perturbation on X\C. Therefore, the labels easily change on y-axis of classified label map with large lambda and the opposite on x-axis. You can think of this as an extension of Section 3 where we have shown that one-pixel attack success rates have checkerboard shape (Figure 2) due to the difference in the number of associated parameters on each pixel. We designed C to be the vulnerable domain which induces attacks on our known C. Thus, we successfully defended attacks with one-pixel padding.
>
> We hope this explanation is clear enough.

---

### Meta-Review · Area_Chair1 · 2018-12-17
**Reject**

**Confidence:** 5
**Recommendation:** Reject

**Metareview:**

The reviewers have agreed this work is not ready for publication at ICLR.